# Exact Post Model Selection Inference for Marginal Screening

**Jason D. Lee**
Computational and Mathematical Engineering
Stanford University
Stanford, CA 94305
jdl17@stanford.edu

**Jonathan E. Taylor**
Department of Statistics
Stanford University
Stanford, CA 94305
jonathan.taylor@stanford.edu

## Abstract

We develop a framework for post model selection inference, via marginal screening, in linear regression. At the core of this framework is a result that characterizes the exact distribution of linear functions of the response $y$, conditional on the model being selected ("condition on selection" framework). This allows us to construct valid confidence intervals and hypothesis tests for regression coefficients that account for the selection procedure. In contrast to recent work in high-dimensional statistics, our results are exact (non-asymptotic) and require no eigenvalue-like assumptions on the design matrix $X$. Furthermore, the computational cost of marginal regression, constructing confidence intervals and hypothesis testing is negligible compared to the cost of linear regression, thus making our methods particularly suitable for extremely large datasets. Although we focus on marginal screening to illustrate the applicability of the condition on selection framework, this framework is much more broadly applicable. We show how to apply the proposed framework to several other selection procedures including orthogonal matching pursuit and marginal screening+Lasso.

## 1 Introduction

Consider the model

$$y_i = \mu(x_i) + \epsilon_i, \ \epsilon_i \sim \mathcal{N}(0, \sigma^2 I), \tag{1}$$

where $\mu(x)$ is an arbitrary function, and $x_i \in \mathbb{R}^p$. Our goal is to perform inference on $(X^T X)^{-1} X^T \mu$, which is the best linear predictor of $\mu$. In the classical setting of $n > p$, the least squares estimator $\hat{\beta} = (X^T X)^{-1} X^T y$ is a commonly used estimator for $(X^T X)^{-1} X^T \mu$. Under the linear model assumption $\mu = X\beta^0$, the exact distribution of $\hat{\beta}$ is

$$\hat{\beta} \sim \mathcal{N}(\beta^0, \sigma^2 (X^T X)^{-1}). \tag{2}$$

Using the normal distribution, we can test the hypothesis $H_0 : \beta_j^0 = 0$ and form confidence intervals for $\beta_j^0$ using the z-test.

However in the high-dimensional $p > n$ setting, the least squares estimator is an underdetermined problem, and the predominant approach is to perform variable selection or model selection [4]. There are many approaches to variable selection including AIC/BIC, greedy algorithms such as forward stepwise regression, orthogonal matching pursuit, and regularization methods such as the Lasso. The focus of this paper will be on the model selection procedure known as marginal screening, which selects the $k$ most correlated features $x_j$ with the response $y$.

Marginal screening is the simplest and most commonly used of the variable selection procedures [9, 21, 16]. Marginal screening requires only $O(np)$ computation and is several orders of magnitude

faster than regularization methods such as the Lasso; it is extremely suitable for extremely large datasets where the Lasso may be computationally intractable to apply. Furthermore, the selection properties are comparable to the Lasso [8].

Since marginal screening utilizes the response variable $y$, the confidence intervals and statistical tests based on the distribution in (2) are not valid; confidence intervals with nominal $1 - \alpha$ coverage may no longer cover at the advertised level:

$$\Pr\left(\beta_j^0 \in C_{1-\alpha}(x)\right) < 1 - \alpha.$$

Several authors have previously noted this problem including recent work in [13, 14, 15, 2]. A major line of work [13, 14, 15] has described the difficulty of inference post model selection: the distribution of post model selection estimates is complicated and cannot be approximated in a uniform sense by their asymptotic counterparts.

In this paper, we describe how to form exact confidence intervals for linear regression coefficients *post model selection*. We assume the model (1), and operate under the fixed design matrix $X$ setting. The linear regression coefficients constrained to a subset of variables $S$ is linear in $\mu$, $e_j^T(X_S^T X_S)^{-1} X_S^T \mu = \eta^T \mu$ for some $\eta$. We derive the conditional distribution of $\eta^T y$ for any vector $\eta$, so we are able to form confidence intervals for regression coefficients.

In Section 2 we discuss related work on high-dimensional statistical inference, and Section 3 introduces the marginal screening algorithm and shows how z intervals may fail to have the correct coverage properties. Section 4 and 5 show how to represent the marginal screening selection event as constraints on $y$, and construct pivotal quantities for the truncated Gaussian. Section 6 uses these tools to develop valid confidence intervals, and Section 7 evaluates the methodology on two real datasets.

Although the focus of this paper is on marginal screening, the "condition on selection" framework, first proposed for the Lasso in [12], is much more general; we use marginal screening as a simple and clean illustration of the applicability of this framework. In Section 8, we discuss several extensions including how to apply the framework to other variable/model selection procedures and to nonlinear regression problems. Section 8 covers 1) marginal screening+Lasso, a screen and clean procedure that first uses marginal screening and cleans with the Lasso, and orthogonal matching pursuit (OMP).

## 2   Related Work

Most of the theoretical work on high-dimensional linear models focuses on *consistency*. Such results establish, under restrictive assumptions on $X$, the Lasso $\hat{\beta}$ is close to the unknown $\beta^0$ [19] and selects the correct model [26, 23, 11]. We refer to the reader to [4] for a comprehensive discussion about the theoretical properties of the Lasso.

There is also recent work on obtaining confidence intervals and significance testing for penalized M-estimators such as the Lasso. One class of methods uses sample splitting or subsampling to obtain confidence intervals and p-values [24, 18]. In the post model selection literature, the recent work of [2] proposed the POSI approach, a correction to the usual t-test confidence intervals by controlling the familywise error rate for all parameters in any possible submodel. The POSI methodology is extremely computationally intensive and currently only applicable for $p \leq 30$.

A separate line of work establishes the asymptotic normality of a corrected estimator obtained by "inverting" the KKT conditions [22, 25, 10]. The corrected estimator $\hat{b}$ has the form $\hat{b} = \hat{\beta} + \lambda\hat{\Theta}\hat{z}$, where $\hat{z}$ is a subgradient of the penalty at $\hat{\beta}$ and $\hat{\Theta}$ is an approximate inverse to the Gram matrix $X^T X$. The two main drawbacks to this approach are 1) the confidence intervals are valid only when the M-estimator is consistent, and thus require restricted eigenvalue conditions on $X$, 2) obtaining $\hat{\Theta}$ is usually much more expensive than obtaining $\hat{\beta}$, and 3) the method is specific to regularized estimators, and does not extend to marginal screening, forward stepwise, and other variable selection methods.

Most closely related to our work is the "condition on selection" framework laid out in [12] for the Lasso. Our work extends this methodology to other variable selection methods such as marginal screening, marginal screening followed by the Lasso (marginal screening+Lasso) and orthogonal matching pursuit. The primary contribution of this work is the observation that many model selection

methods, including marginal screening and Lasso, lead to "selection events" that can be represented as a set of constraints on the response variable $y$. By conditioning on the selection event, we can characterize the exact distribution of $\eta^T y$. This paper focuses on marginal screening, since it is the simplest of variable selection methods, and thus the applicability of the "condition on selection event" framework is most transparent. However, this framework is not limited to marginal screening and can be applied to a wide a class of model selection procedures including greedy algorithms such as orthogonal matching pursuit. We discuss some of these possible extensions in Section 8, but leave a thorough investigation to future work.

A remarkable aspect of our work is that we only assume $X$ is in general position, and the test is exact, meaning the distributional results are true even under finite samples. By extension, we do not make any assumptions on $n$ and $p$, which is unusual in high-dimensional statistics [4]. Furthermore, the computational requirements of our test are negligible compared to computing the linear regression coefficients.

## 3 Marginal Screening

Let $X \in \mathbb{R}^{n \times p}$ be the design matrix, $y \in \mathbb{R}^n$ the response variable, and assume the model $y_i = \mu(x_i) + \epsilon_i, \epsilon_i \sim \mathcal{N}(0, \sigma^2 I)$. We will assume that $X$ is in general position and has unit norm columns. The algorithm estimates $\hat{\beta}$ via Algorithm 1. The marginal screening algorithm chooses

---
**Algorithm 1** Marginal screening algorithm
---
1: **Input:** Design matrix $X$, response $y$, and model size $k$.
2: Compute $|X^T y|$.
3: Let $\hat{S}$ be the index of the $k$ largest entries of $|X^T y|$.
4: Compute $\hat{\beta}_{\hat{S}} = (X_{\hat{S}}^T X_{\hat{S}})^{-1} X_{\hat{S}}^T y$

---

the $k$ variables with highest absolute dot product with $y$, and then fits a linear model over those $k$ variables. We will assume $k \leq \min(n, p)$. For any fixed subset of variables $S$, the distribution of $\hat{\beta}_S = (X_S^T X_S)^{-1} X_S^T y$ is

$$\hat{\beta}_S \sim \mathcal{N}\left((X_S^T X_S)^{-1} X_S^T \mu, \sigma^2 (X_S^T X_S)^{-1}\right) \tag{3}$$

We will use the notation $\beta^\star_{j \in S} := (\beta^\star_S)_j$, where $j$ is indexing a variable in the set $S$. The $z$-test intervals for a regression coefficient are

$$C(\alpha, j, S) := \left(\hat{\beta}_{j \in S} - \sigma z_{1-\alpha/2}(X_S^T X_S)_{jj}, \hat{\beta}_{j \in S} + \sigma z_{1-\alpha/2}(X_S^T X_S)_{jj}\right) \tag{4}$$

and each interval has $1 - \alpha$ coverage, meaning $\Pr\left(\beta^\star_{j \in S} \in C(\alpha, j, S)\right) = 1 - \alpha$. However if $\hat{S}$ is chosen using a model selection procedure that depends on $y$, the distributional result (3) no longer holds and the z-test intervals will not cover at the $1 - \alpha$ level, and $\Pr\left(\beta^\star_{j \in \hat{S}} \in C(\alpha, j, \hat{S})\right) < 1 - \alpha$.

### 3.1 Failure of z-test confidence intervals

We will illustrate empirically that the z-test intervals do not cover at $1 - \alpha$ when $\hat{S}$ is chosen by marginal screening in Algorithm 1. For this experiment we generated $X$ from a standard normal with $n = 20$ and $p = 200$. The signal vector is 2 sparse with $\beta_1^0, \beta_2^0 = $ SNR, $y = X\beta^0 + \epsilon$, and $\epsilon \sim N(0, 1)$. The confidence intervals were constructed for the $k = 2$ variables selected by the marginal screening algorithm. The $z$-test intervals were constructed via (4) with $\alpha = .1$, and the adjusted intervals were constructed using Algorithm 2. The results are described in Figure 1.

## 4 Representing the selection event

Since Equation (3) does not hold for a selected $\hat{S}$ when the selection procedure depends on $y$, the $z$-test intervals are not valid. Our strategy will be to understand the conditional distribution of $y$

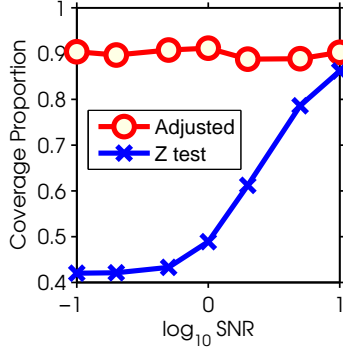

Figure 1: Plots of the coverage proportion across a range of SNR (log-scale). We see that the coverage proportion of the z intervals can be far below the nominal level of $1 - \alpha = .9$, even at SNR =5. The adjusted intervals always have coverage proportion .9. Each point represents 500 independent trials.

and contrasts (linear functions of $y$) $\eta^T y$, then construct inference conditional on the selection event $\hat{E}$. We will use $\hat{E}(y)$ to represent a random variable, and $E$ to represent an element of the range of $\hat{E}(y)$. In the case of marginal screening, the selection event $\hat{E}(y)$ corresponds to the set of selected variables $\hat{S}$ and signs $s$:

$$
\begin{aligned}
\hat{E}(y) &= \left\{ y : \text{sign}(x_i^T y) x_i^T y > \pm x_j^T y \text{ for all } i \in \hat{S} \text{ and } j \in \hat{S}^c \right\} \\
&= \left\{ y : \hat{s}_i x_i^T y > \pm x_j^T y \text{ and } \hat{s}_i x_i^T y \geq 0 \text{ for all } i \in \hat{S} \text{ and } j \in \hat{S}^c \right\} \\
&= \left\{ y : A(\hat{S}, \hat{s}) y \leq b(\hat{S}, \hat{s}) \right\}
\end{aligned}
\tag{5}
$$

for some matrix $A(\hat{S}, \hat{s})$ and vector $b(\hat{S}, \hat{s})$[1]. We will use the selection event $\hat{E}$ and the selected variables/signs pair $(\hat{S}, \hat{s})$ interchangeably since they are in bijection.

The space $\mathbb{R}^n$ is partitioned by the selection events, $\mathbb{R}^n = \bigsqcup_{(S,s)} \{y : A(S,s)y \leq b(S,s)\}$[2]. The vector $y$ can be decomposed with respect to the partition as follows $y = \sum_{S,s} y \, \mathbf{1}\,(A(S,s)y \leq b(S,s))$.

**Theorem 4.1.** *The distribution of $y$ conditional on the selection event is a constrained Gaussian,*

$$
y \,|\, \{\hat{E}(y) = E\} \stackrel{d}{=} z \,|\, \{A(S,s)z \leq b\}, \; z \sim \mathcal{N}(\mu, \sigma^2 I).
$$

*Proof.* The event $E$ is in bijection with a pair $(S, s)$, and $y$ is unconditionally Gaussian. Thus the conditional $y \,|\, \{A(S,s)y \leq b(S,s)\}$ is a Gaussian constrained to the set $\{A(S,s)y \leq b(S,s)\}$. $\square$

## 5 Truncated Gaussian test

This section summarizes the recent tools developed in [12] for testing contrasts[3] $\eta^T y$ of a constrained Gaussian $y$. The results are stated without proof and the proofs can be found in [12]. The primary result is a one-dimensional pivotal quantity for $\eta^T \mu$. This pivot relies on characterizing the distribution of $\eta^T y$ as a truncated normal. The key step to deriving this pivot is the following lemma:

**Lemma 5.1.** *The conditioning set can be rewritten in terms of $\eta^T y$ as follows:*

$$
\{Ay \leq b\} = \{\mathcal{V}^-(y) \leq \eta^T y \leq \mathcal{V}^+(y), \mathcal{V}^0(y) \geq 0\}
$$

*where*

$$\alpha = \frac{A\Sigma\eta}{\eta^T\Sigma\eta} \tag{6}$$

$$\mathcal{V}^- = \mathcal{V}^-(y) = \max_{j:\,\alpha_j<0} \frac{b_j - (Ay)_j + \alpha_j\eta^T y}{\alpha_j} \tag{7}$$

$$\mathcal{V}^+ = \mathcal{V}^+(y) = \min_{j:\,\alpha_j>0} \frac{b_j - (Ay)_j + \alpha_j\eta^T y}{\alpha_j}. \tag{8}$$

$$\mathcal{V}^0 = \mathcal{V}^0(y) = \min_{j:\,\alpha_j=0} b_j - (Ay)_j \tag{9}$$

*Moreover, $(\mathcal{V}^+, \mathcal{V}^-, \mathcal{V}^0)$ are independent of $\eta^T y$.*

**Theorem 5.2.** *Let $\Phi(x)$ denote the CDF of a $N(0,1)$ random variable, and let $F^{[a,b]}_{\mu,\sigma^2}$ denote the CDF of $TN(\mu,\sigma,a,b)$, i.e.:*

$$F^{[a,b]}_{\mu,\sigma^2}(x) = \frac{\Phi((x-\mu)/\sigma) - \Phi((a-\mu)/\sigma)}{\Phi((b-\mu)/\sigma) - \Phi((a-\mu)/\sigma)}. \tag{10}$$

*Then $F^{[\mathcal{V}^-,\mathcal{V}^+]}_{\eta^T\mu,\,\eta^T\Sigma\eta}(\eta^T y)$ is a pivotal quantity, conditional on $\{Ay \le b\}$:*

$$F^{[\mathcal{V}^-,\mathcal{V}^+]}_{\eta^T\mu,\,\eta^T\Sigma\eta}(\eta^T y) \,\big|\, \{Ay \le b\} \sim \mathrm{Unif}(0,1) \tag{11}$$

*where $\mathcal{V}^-$ and $\mathcal{V}^+$ are defined in* (7) *and* (8).

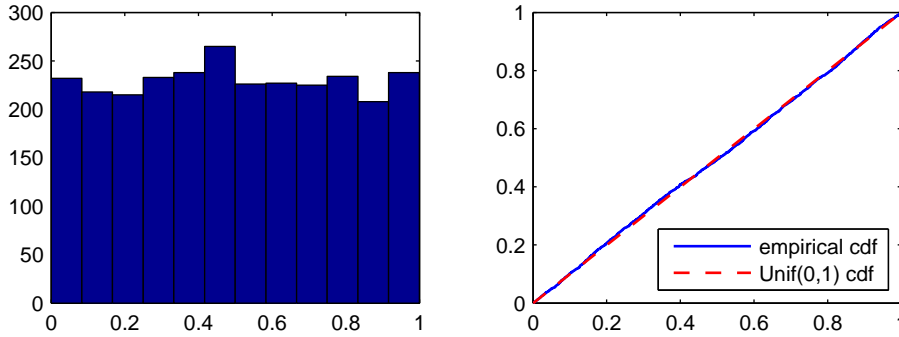

Figure 2: Histogram and qq plot of $F^{[\mathcal{V}^-,\mathcal{V}^+]}_{\eta^T\mu,\,\eta^T\Sigma\eta}(\eta^T y)$ where $y$ is a constrained Gaussian. The distribution is very close to $\mathrm{Unif}(0,1)$, which is in agreement with Theorem 5.2.

# 6 Inference for marginal screening

In this section, we apply the theory summarized in Sections 4 and 5 to marginal screening. In particular, we will construct confidence intervals for the selected variables.

To summarize the developments so far, recall that our model (1) says that $y \sim N(\mu, \sigma^2 I)$. The distribution of interest is $y|\{\hat{E}(y) = E\}$, and by Theorem 4.1, this is equivalent to $y|\{A(S,s)z \le b(S,s)\}$, where $y \sim N(\mu, \sigma^2 I)$. By applying Theorem 5.2, we obtain the pivotal quantity

$$F^{[\mathcal{V}^-,\mathcal{V}^+]}_{\eta^T\mu,\,\sigma^2||\eta||_2^2}(\eta^T y) \,\big|\, \{\hat{E}(y) = E\} \sim \mathrm{Unif}(0,1) \tag{12}$$

for any $\eta$, where $\mathcal{V}^-$ and $\mathcal{V}^+$ are defined in (7) and (8).

In this section, we describe how to form confidence intervals for the components of $\beta^\star_{\hat{S}} = (X^T_{\hat{S}}X_{\hat{S}})^{-1}X^T_{\hat{S}}\mu$. The best linear predictor of $\mu$ that uses only the selected variables is $\beta^\star_{\hat{S}}$, and $\hat{\beta}_{\hat{S}} = (X^T_{\hat{S}}X_{\hat{S}})^{-1}X^T_{\hat{S}}y$ is an unbiased estimate of $\beta^\star_{\hat{S}}$. If we choose

$$\eta_j = ((X^T_{\hat{S}}X_{\hat{S}})^{-1}X^T_{\hat{S}}e_j)^T, \tag{13}$$

then $\eta_j^T \mu = \beta^\star_{j \in \hat{S}}$, so the above framework provides a method for inference about the $j^{\text{th}}$ variable in the model $\hat{S}$.

## 6.1 Confidence intervals for selected variables

Next, we discuss how to obtain confidence intervals for $\beta^\star_{j \in \hat{S}}$. The standard way to obtain an interval is to invert a pivotal quantity [5]. In other words, since $\mathbf{Pr}\left( \frac{\alpha}{2} \leq F^{[\mathcal{V}^-, \mathcal{V}^+]}_{\beta^\star_{j \in \hat{S}}, \, \sigma^2 ||\eta_j||^2}(\eta_j^T y) \leq 1 - \frac{\alpha}{2} \,\big|\, \{\hat{E} = E\} \right) = \alpha$ one can define a $(1 - \alpha)$ (conditional) confidence interval for $\beta^\star_{j, \hat{E}}$ as

$$\left\{ x : \frac{\alpha}{2} \leq F^{[\mathcal{V}^-, \mathcal{V}^+]}_{x, \, \sigma^2 ||\eta_j||^2}(\eta_j^T y) \leq 1 - \frac{\alpha}{2} \right\}. \tag{14}$$

In fact, $F$ is monotone decreasing in $x$, so to find its endpoints, one need only solve for the root of a smooth one-dimensional function. The monotonicity is a consequence of the fact that the truncated Gaussian distribution is a natural exponential family and hence has monotone likelihood ratio in $\mu$ [17].

We now formalize the above observations in the following result, an immediate consequence of Theorem 5.2.

**Corollary 6.1.** *Let $\eta_j$ be defined as in* (13)*, and let $L_\alpha = L_\alpha(\eta_j, (\hat{S}, \hat{s}))$ and $U_\alpha = U_\alpha(\eta_j, (\hat{S}, \hat{s}))$ be the (unique) values satisfying*

$$F^{[\mathcal{V}^-, \mathcal{V}^+]}_{L_\alpha, \, \sigma^2 ||\eta_j||^2}(\eta_j^T y) = 1 - \frac{\alpha}{2} \qquad\qquad F^{[\mathcal{V}^-, \mathcal{V}^+]}_{U_\alpha, \, \sigma^2 ||\eta_j||^2}(\eta_j^T y) = \frac{\alpha}{2} \tag{15}$$

*Then $[L_\alpha, U_\alpha]$ is a $(1 - \alpha)$ confidence interval for $\beta^\star_{j \in \hat{S}}$, conditional on $\hat{E}$:*

$$\mathbb{P}\left( \beta^\star_{j \in \hat{S}} \in [L_\alpha, U_\alpha] \,\big|\, \{\hat{E} = E\} \right) = 1 - \alpha. \tag{16}$$

*Proof.* The confidence region of $\beta^\star_{j \in \hat{S}}$ is the set of $\beta_j$ such that the test of $H_0 : \beta^\star_{j \in \hat{S}}$ accepts at the $1 - \alpha$ level. The function $F^{[\mathcal{V}^-, \mathcal{V}^+]}_{x, \, \sigma^2 ||\eta_j||^2}(\eta_j^T y)$ is monotone in $x$, so solving for $L_\alpha$ and $U_\alpha$ identify the most extreme values where $H_0$ is still accepted. This gives a $1 - \alpha$ confidence interval. $\square$

Next, we establish the unconditional coverage of the constructed confidence intervals and the false coverage rate (FCR) control [1].

**Corollary 6.2.** *For each $j \in \hat{S}$,*

$$\mathbf{Pr}\left( \beta^\star_{j \in \hat{S}} \in [L^j_\alpha, U^j_\alpha] \right) = 1 - \alpha. \tag{17}$$

*Furthermore, the FCR of the intervals $\left\{ [L^j_\alpha, U^j_\alpha] \right\}_{j \in \hat{E}}$ is $\alpha$.*

*Proof.* By (16), the conditional coverage of the confidence intervals are $1 - \alpha$. The coverage holds for every element of the partition $\{\hat{E}(y) = E\}$, so

$$\mathbf{Pr}\left( \beta^\star_{j \in \hat{S}} \in [L^j_\alpha, U^j_\alpha] \right) = \sum_E \mathbf{Pr}\left( \beta^\star_{j \in \hat{S}} \in [L_\alpha, U_\alpha] \,\big|\, \{\hat{E} = E\} \right) \mathbf{Pr}(\hat{E} = E)$$

$$= \sum_E (1 - \alpha)\, \mathbf{Pr}(\hat{E} = E) = 1 - \alpha.$$

$\square$

**Remark 6.3.** *We would like to emphasize that the previous Corollary shows that the constructed confidence intervals are unconditionally valid. The conditioning on the selection event $\hat{E}(y) = E$ was only for mathematical convenience to work out the exact pivot. Unlike standard $z$-test intervals, the coverage target, $\beta^\star_{j \in \hat{S}}$, and the interval $[L_\alpha, U_\alpha]$ are random. In a typical confidence interval only the interval is random; however in the post-selection inference setting, the selected model is random, so both the interval and the target are necessarily random [2].*

We summarize the algorithm for selecting and constructing confidence intervals below.

**Algorithm 2** Confidence intervals for selected variables

1: **Input:** Design matrix $X$, response $y$, model size $k$.
2: Use Algorithm 1 to select a subset of variables $\hat{S}$ and signs $\hat{s} = \text{sign}(X_{\hat{S}}^T y)$.
3: Let $A = A(\hat{S}, \hat{s})$ and $b = b(\hat{S}, \hat{s})$ using (5). Let $\eta_j = (X_{\hat{S}}^T)^{\dagger} e_j$.
4: Solve for $L_{\alpha}^j$ and $U_{\alpha}^j$ using Equation (15) where $\mathcal{V}^-$ and $\mathcal{V}^+$ are computed via (7) and (8) using the $A$, $b$, and $\eta_j$ previously defined.
5: **Output:** Return the intervals $[L_{\alpha}^j, U_{\alpha}^j]$ for $j \in \hat{S}$.

## 7 Experiments

In Figure 1, we have already seen that the confidence intervals constructed using Algorithm 2 have exactly $1 - \alpha$ coverage proportion. In this section, we perform two experiments on real data where the linear model does not hold, the noise is not Gaussian, and the noise variance is unknown.

### 7.1 Diabetes dataset

The diabetes dataset contains $n = 442$ diabetes patients measured on $p = 10$ baseline variables [6]. The baseline variables are age, sex, body mass index, average blood pressure, and six blood serum measurements, and the response $y$ is a quantitative measure of disease progression measured one year after the baseline. Since the noise variance $\sigma^2$ is unknown, we estimate it by $\sigma^2 = \frac{\|y - \hat{y}\|}{n - p}$,

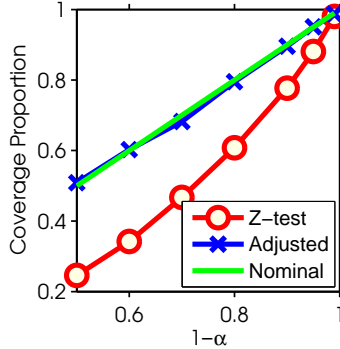

Figure 3: Plot of $1 - \alpha$ vs the coverage proportion for diabetes dataset. The nominal curve is the line $y = x$. The coverage proportion of the adjusted intervals agree with the nominal coverage level, but the z-test coverage proportion is strictly below the nominal level. The adjusted intervals perform well, despite the noise being non-Gaussian, and $\sigma^2$ unknown.

where $\hat{y} = X\hat{\beta}$ and $\hat{\beta} = (X^T X)^{-1} X^T y$. For each trial we generated new responses $\tilde{y}_i = X\hat{\beta} + \tilde{\epsilon}$, and $\tilde{\epsilon}$ is bootstrapped from the residuals $r_i = y_i - \hat{y}_i$. We used marginal screening to select $k = 2$ variables, and then fit linear regression on the selected variables. The adjusted confidence intervals were constructed using Algorithm 2 with the estimated $\sigma^2$. The nominal coverage level is varied across $1 - \alpha \in \{.5, .6, .7, .8, .9, .95, .99\}$. From Figure 3, we observe that the adjusted intervals always cover at the nominal level, whereas the z-test is always below. The experiment was repeated 2000 times.

### 7.2 Riboflavin dataset

Our second data example is a high-throughput genomic dataset about riboflavin (vitamin B2) production rate [3]. There are $p = 4088$ variables which measure the log expression level of different genes, a single real-valued response $y$ which measures the logarithm of the riboflavin production rate, and $n = 71$ samples. We first estimate $\sigma^2$ by using cross-validation [20], and apply marginal screening with $k = 30$, as chosen in [3]. We then use Algorithm 2 to identify genes significant at

$\alpha = 10\%$. The genes identified as significant were YCKE_at, YOAB_at, and YURQ_at. After using Bonferroni to control for FWER, we found YOAB_at remained significant.

# 8 Extensions

The purpose of this section is to illustrate the broad applicability of the condition on selection framework. For expository purposes, we focused the paper on marginal screening where the framework is particularly easy to understand. In the rest of this section, we show how to apply the framework to marginal screening+Lasso, and orthogonal matching pursuit. This is a non-exhaustive list of selection procedures where the condition on selection framework is applicable, but we hope this incomplete list emphasizes the ease of constructing tests and confidence intervals post-model selection via conditioning.

## 8.1 Marginal screening + Lasso

The marginal screening+Lasso procedure was introduced in [7] as a variable selection method for the ultra-high dimensional setting of $p = O(e^{n^k})$. Fan et al. [7] recommend applying the marginal screening algorithm with $k = n - 1$, followed by the Lasso on the selected variables. This is a two-stage procedure, so to properly account for the selection we must encode the selection event of marginal screening followed by Lasso. This can be done by representing the two stage selection as a single event. Let $(\hat{S}_m, \hat{s}_m)$ be the variables and signs selected by marginal screening, and the $(\hat{S}_L, \hat{z}_L)$ be the variables and signs selected by Lasso [12]. In Proposition 2.2 of [12], it is shown how to encode the Lasso selection event $(\hat{S}_L, \hat{z}_L)$ as a set of constraints $\{A_L y \le b_L\}$ [4], and in Section 4 we showed how to encode the marginal screening selection event $(\hat{S}_m, \hat{s}_m)$ as a set of constraints $\{A_m y \le b_m\}$. Thus the selection event of marginal screening+Lasso can be encoded as $\{A_L y \le b_L, A_m y \le b_m\}$. Using these constraints, the hypothesis test and confidence intervals described in Algorithm 2 are valid for marginal screening+Lasso.

## 8.2 Orthogonal Matching Pursuit

Orthogonal matching pursuit (OMP) is a commonly used variable selection method. At each iteration, OMP selects the variable most correlated with the residual $r$, and then recomputes the residual using the residual of least squares using the selected variables. Similar to Section 4, we can represent the OMP selection event as a set of linear constraints on $y$.

$$
\begin{aligned}
\hat{E}(y) &= \{y : \operatorname{sign}(x_{p_i}^T r_i) x_{p_i}^T r_i > \pm x_j^T r_i, \text{ for all } j \ne p_i \text{ and all } i \in [k]\} \\
&= \{y : \hat{s}_i x_{p_i}^T (I - X_{\hat{S}_{i-1}} X_{\hat{S}_{i-1}}^\dagger) y > \pm x_j^T (I - X_{\hat{S}_{i-1}} X_{\hat{S}_{i-1}}^\dagger) y \text{ and} \\
&\qquad \hat{s}_i x_{p_i}^T (I - X_{\hat{S}_{i-1}} X_{\hat{S}_{i-1}}^\dagger) y > 0, \text{ for all } j \ne p_i, \text{ and all } i \in [k] \}
\end{aligned}
$$

The selection event encodes that OMP selected a certain variable and the sign of the correlation of that variable with the residual, at steps 1 to $k$. The primary difference between the OMP selection event and the marginal screening selection event is that the OMP event also describes the order at which the variables were chosen.

# 9 Conclusion

Due to the increasing size of datasets, marginal screening has become an important method for fast variable selection. However, the standard hypothesis tests and confidence intervals used in linear regression are invalid after using marginal screening to select important variables. We have described a method to form confidence intervals after marginal screening. The condition on selection framework is not restricted to marginal screening, and also applies to OMP and marginal screening + Lasso. The supplementary material also discusses the framework applied to non-negative least squares.

## Footnotes

[1]$b$ can be taken to be 0 for marginal screening, but this extra generality is needed for other model selection methods.

[2]It is also possible to use a coarser partition, where each element of the partition only corresponds to a subset of variables $S$. See [12] for details.

[3]A contrast of $y$ is a linear function of the form $\eta^T y$.

[4]The Lasso selection event is with respect to the Lasso optimization problem after marginal screening.

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
