[Reviews · NeurIPS 2014]

Submitted by Assigned_Reviewer_28

This paper presents a framework for post model selection inference in the context of marginal screening, which is a computationally efficient way of variable selection in high dimensional linear regression problems. While the paper is focused on marginal screening, the approach is applicable to a broad range of problems.

The paper is well written, very clear. I am not an expert in this area, but it seems like a significant problem and a statistically solid approach.

Small comments: the paper may be more accessible if the authors add more intuition before some of the equations. For example the definition of \hat{E}_y on line 185 is confusing with the plus/minus symbol; it may be worth it to point out that this merely translates to picking the k largest values in X^Ty.

Line 157 speaks of Algorithm 2 for the first time. The authors should point out that this is the improved algorithm.

Often \mu and y are used interchangeably, this needs to be clarified.

Lemma 5.1: Is Sigma the covariance matrix of y? There is no definition of Sigma in the statement.

Sec 7.2: Its not clear whats the point of this section. The authors report some genes, but give no explanation of why that is an interesting result.
Summary: I think the paper has an statistically sound and computationally efficient approach to an important problem.

Submitted by Assigned_Reviewer_31

The purpose of this paper is to derive exact confidence intervals for linear regression coefficients post model selection. The authors first introduce the notation of the regression model under study and the marginal screening as one simple algorithm for variable selection. They illustrate empirically that the commonly used z-test interval is not correctly calibrated and thus not valid. The authors then construct the distribution of the target variable conditioned on the selection event, which leads to confidence intervals for, e.g., the marginal screening. Finally, the confidence intervals are studied empirically on two real-world datasets.

* Quality:

Overall the paper is technically sound. However, it fails to evaluate all strength and weaknesses of the proposed method. Specifically:
- In hypothesis testing (confidence intervals), there is always the trade-off between type I and II error. The type I error is, unfortunately, not well-studied. This could be done, e.g., with simulated data where you have control over the ground truths (correlation between features and target label).
- You state that the POSI [2] method is computational expensive and thus not applicable for d larger than 30. But that means that you can use POSI in Section 7.1 or study its behaviour on artificial data.
- There are missing baselines: What is the performance (power and correctness) compared to other baselines as greedy algorithms, POSI,... mentioned in Section 2?
- The depending on the number of dimensions and the coverage rate depending on n would also be worth to be studied.

* Clarity:

Overall, the paper is comprehensibly structured, but the context is often rarely described, which makes it hard to follow. Furthermore, some key concepts and notation are ambiguous or not introduced. Specifically,
- The model is introduced immediately in the first sentence but the notation is actually formally defined in Section 3. At the first part some aspects are incompletely presented. E.g., what is the number of examples? What is \mu (in contrast to \mu(x))?
- The notation for the observation and the noise free label is confusing: "(X^TX)^{−1} X^T\mu, which is the best linear predictor of \mu" (line 039) makes not really sense to me. Isn't \mu the best estimator for \mu? Do you mean "(X^TX)^{−1} X^T y" is the best predictor of \mu? The same holds for "\mu = X \beta^0".
- \mu(x) is assumed to be arbitrary in the beginning but only the linear case is studied. To clarify things, why don't you restrict yourself to linear models at all?
- While the need for variable selection is motivated the need of confidence intervals is not. Can you clarify this in more detail?
- 7.1 studies the dependance between empirical and theoretical coverage. However, Section 7.2 present only one outcome of your method on the genomic dataset. What is the significance of that statement? Is that already known and does it confirm the correctness of your approach or is that a new outcome and gives new insight to the community?
- "Z-test", "Adjusted", and "Nominal" in Figure 3 not introduced

* Originality:

The paper applies the idea of [12] to derive the exact distribution of the target variables by conditioning on the selection event to marginal screening.

* Significance:

The empirical study fails to evaluate all strength and weaknesses and the theoretical results are quite similar to [12].
Summary: + interesting idea
- minor contribution, since the approach is quite similar to [12]
- weak experiments
- the paper suffers from its presentation; the significance of the results are often not clear and some details are confusing

Submitted by Assigned_Reviewer_37

The paper deals with CI for the regression coefficients after selection by marginal regression. The Authors construct a pivot for the distribution of a regression coefficient.
The main result seems to be in Corollary 6.1 & 6.2. The notation is not that trivial to understand, but I'm not sure how to read it. To be valid, j in this corollaries should be fixed, but \hat S is random. So what do the corollaries say? If j is random, you should be penalized for the multiplicity. But can it be fixed in advance when the set is random? Also. The main interest in CI is for those coefficient that are found to be 0. For the large coefficients the main contribution to the error is the bias due to the selection.
Summary: The approach is refreshing and nice. I'm not sure about its application.
Author Feedback
Author rebuttal: We thank the reviewers for their careful reading of our manuscript. The primary contribution of our paper is to extend the tools of [12] from the lasso to for a large class of variable selection procedures including marginal screening, forward stepwise regression/orthogonal matching pursuit, non-negative least squares, and isotonic regression. We emphasized marginal screening, since it is the simplest to describe and the most commonly used in practice. Our response is focused on a few specific concerns raised in Review 2, and the exposition and presentation that Review 1 and 3 commented on. Both Reviews 1 and 2 commented on the definition of \mu, so we address this first. We apologize for the confusion.
\mu(x) and y:
1. y_i=\mu(x_i) +\epsilon, where \mu(x) is an arbitrary nonlinear function. Thus \mu(x_i)= E(y |x=x_i). We do not assume the linear model y =x^T\beta^0 +\epsilon, since this is rarely the case. Even though the true data generating distribution is not linear, the statistician still frequently performs linear regression especially in the p>n regime, where the number of data points is insufficient to fit a nonparametric model. The focus of our work is on linear regression after variable selection. Our results are applicable even when the linear model is misspecified. We consider this
2. The vector \mu=[\mu(x_1),…,\mu(x_n)] is the function \mu evaluated at the design points.
3. \beta_{E} = (X_S^T X_S)^{-1} X_S^T \mu is the best linear predictor of \mu using only the variables in E. In line 39, we were commenting on the traditional case of n>p, so the quantity \beta =(X^TX)^-1 X^T \mu is well-defined. This is the best linear predictor of the nonlinear \mu.

Review 1 (assigned_reviewer_28):
4. We agree that the definition of \hat{E}_y on line 185 is confusing. We will point out that it is choosing the largest elements of X^T y.
5. Yes, \Sigma is the covariance of y. In the usual case of regression where errors are independent, Sigma =\sigma^2 I. We will clarify this. Thanks!
6. We agree that this section does not draw any scientific conclusions. The only point was to illustrate on a real dataset with p>>n. The dataset was only released in early 2014, and we are unaware of any experimental studies that have declared certain genes significant in riboflavin production rate.
Review 2 (Assigned_Reviewer_31)
1. Type 1 error. Our method controls the type 1 error at exactly \alpha, when the noise is Gaussian and variance is known. In 7.1 we empirically demonstrated that when the noise distribution is not Gaussian, and the variance is estimated, the type 1 error is still controlled at the nominal level of \alpha. Figures 1,2, and 3 show that the type 1 error is controlled at alpha.
2. POSI. The POSI paper did not have experiments/code, so we did not compare against them. The POSI intervals are provably between sqrt(2\log p ) to sqrt(p) times longer than the usual Gaussian z-test intervals. For the diabetes dataset, this means they are between 2.14 to 3.16 times longer. Our method generates intervals on average 2.1 times longer for this dataset. We expect the difference to be more drastic for larger p, since the POSI constants scale as sqrt(p). Also, the method is exponential in p, which makes it unsuitable for realistic sizes.
3. The KKT inverting method is only applicable to regularized estimators such as lasso. We are not aware of how to apply these to marginal screening or OMP. The sample splitting techniques are also not comparable because they tend to choose different models than performing marginal screening on the full dataset. It is easy to prove that our procedure always has more power, when we use the uniformly most accurate intervals.
4. Our type 1 error is always controlled at alpha. This is completely independent of n and p.
5. In Figure 4, “z-test” are the standard Gaussian intervals, “Nominal” is the nominal coverage level of 1-\alpha, and “adjusted” is our procedure. We will clarify this in the caption.
6. Our primary contribution is to generalize [12] to other variable selection algorithms. From [12], it is not apparent that the procedure can be used beyond the lasso.
7. Confidence intervals are required to evaluate whether any selected variable is significant. A variable that is selected by a variable selection procedure need not be statistically significant, which is why we need confidence intervals to assess the uncertainty in the estimates. For example running a variable selection procedure on noise would return some subset of variables, but they are not significant.
Review 3 (assigned_reviewer_37)
1. The two Corollaries state that the type 1 error of the confidence intervals is alpha. The confidence intervals are marginal confidence intervals, like in the usual Gaussian z-test intervals, so they hold for any single j. The correction for multiplicity can be done by Bonferroni, as done in the usual Gaussian z-test intervals. It would also be possible to construct simultaneous intervals, but these would not be elliptically shaped as in the usual Gaussian case.
2. The estimator \hat \beta_{\hat \E} is not unbiased for \beta^\star _{\E}, but it is the least squares after variable selection estimate. An approximately unbiased estimate can be found by performing MLE restricted to the truncated normal. We will add this to the paper.